



# Terrain visibility impact on the preparation of landslide inventories: some practical cases

Txomin Bornaetxea[1], Ivan Marchesini[2], Sumit Kumar[3], Rabisankar Karmakar[3], Alessandro Mondini[2]

[1]Euskal Herriko Unibertsitatea (UPV/EHU), Barrio Sarriena s/n, 48940 Leioa, Spain
[2]CNR-IRPI, via Madonna Alta 126, 06128 Perugia, Italy
[3]Geohazard Research and Management Centre, Geological Survey of India, Kolkata, India

*Correspondence to*: Txomin Bornaetxea (txomin.bornaetxea@ehu.eus)

**Abstract.** Landslide inventories are used for multiple purposes including landscape characterisation and monitoring, and landslide susceptibility, hazard and risk evaluation. Their quality can depend on the data and the methods with which they were produced. In this work we evaluate the effects of a variable visibility of the territory to map on the spatial distribution of the information collected by four landslide inventories prepared using different approaches in two study areas.

The method first classifies the territory in areas with different visibility levels from the paths (roads) used to map landslides, and then estimates the landslide density reported in the inventories into the different visibility classes.

Our results show that 1) the density of the information is strongly related to the visibility in inventories obtained through fieldwork, technical reports and/or newspapers, where landslides are under-sampled in low visibility classes; and 2) the inventories obtained by photo-interpretation of images suffer from a marked under representation of small landslides close to roads or infrastructures. We maintain that the proposed procedure can be useful to evaluate the quality of landslide inventories and then properly orient their use.

## 1 Introduction

Landslides affect the evolution of the territory and represent a hazard to the population, structures and infrastructure (Fell et al., 2008). Detailed information about the spatial and temporal distribution, and characteristics of past landslides is essential for susceptibility/hazard statistical (Hao et al., 2020; Reichenbach et al., 2018; Steger et al., 2016; Van Den Eeckhaut and Hervás, 2012) and physically-based modelling (Lee et al., 2020; Park et al., 2019).

Complete landslide inventories are difficult or impossible to achieve (Corominas et al., 2014) and when they are used, they should at least be statistically representative of the slope processes occurring in the studied area (Cova et al., 2018; Guzzetti et al., 2012; Melzner et al., 2020).

Bias in sampling can prevent the realization of statistically representative inventories and introduce errors that are difficult to manage, propagate and communicate (Guzzetti et al., 1999). Largely incomplete landslide inventories can have relevant impact on derivatives products such as landslide susceptibility and hazard maps (Steger et al., 2021, 2017).

Differences in completeness and then quality of landslide inventories can largely depend on the mapping approach, the study area extent or the analysed time span, the availability of data, time and human resources (Fiorucci et al., 2018; Mondini et al., 2014; Santangelo et al., 2015). Inventories can be compiled in several ways (Guzzetti et al., 2012) and exploiting different sources of data. In the case of non-automatic (or 'manual') methods which include visual interpretation of remote sensing

images, direct field investigation and reorganisation of data inherited from historical archives a good visibility of the territory can become a key factor (Bornaetxea and Marchesini, 2021; Steger et al., 2021).

In this paper we investigate if and how the terrain visibility can limit the mapping capacity of an operator, and then influence the quality of an inventory. In other words, we compare the "data collection effect" (Steger et al., 2021) produced by the mapping of landslides in the field, with that determined by the recognition of landslides by photo interpretation of remote

sensing images. In fact, Steger et al (2021) argued that the spatial distribution of landslides also depends on the "effects" generated by the adopted data collection procedure.

For this purpose, we studied a purely field-based landslide inventory available for the Gipuzkoa Province (Spain) (Bornaetxea et al., 2018), and a few inventories covering the Darjeeling district (north-east of India), obtained by interpreting several types of remotely sensed data, conducting field surveys from roads and collecting other types of information. The paper is organized

as follows. In Sec. 2 we discuss the rationale behind the research. In Sec. 3 we summarize the method of the analysis while in Secs. 4 and 5 we describe the study area and the data used. Section 6 describes the visibility maps of our study areas and Sec. 7 shows the results. In Sec. 8 we discuss findings, and we draw conclusions in Sec. 9.

## 2 Rationale

It is widely accepted the primary role that landslide inventories play for (i) showing the location and type of landslides in a

region, (ii) mapping the effects of landslide triggering events, (iii) describing the abundance of mass movements, (iv) determining the frequency-area statistics of slope failures, and (v) providing relevant information to train and validate landslide susceptibility and/or hazard models (Galli et al., 2008; Guzzetti et al., 2012). The usefulness of a landslide susceptibility map is directly related to the quality of the data used to build the model (Cascini, 2008; Corominas et al., 2014; Fressard et al., 2014; Guzzetti et al., 2006; van Westen et al., 2008). The propagation of the error caused by large incompleteness in the

inventories used to produce a susceptibility map was investigated by Steger et al. (2016) and Steger et al. (2017) in Lower Austria. They discovered that biased input data can generate unrealistic (or even meaningless) results, and enhance the apparent predictive performance of a model.

Quality requirements depend on the inventory usage. Geographical accuracy (Santangelo et al., 2015) and representativeness are relevant for susceptibility analysis when carried out by means of statistical models (Steger et al., 2021), while occurrence

dates, size and location are prioritized for damage evaluation studies, also related to climate changes (Gariano and Guzzetti, 2016).



According to Guzzetti et al. (2012) the quality of a landslide inventory refers to geographical and thematic information accuracy, in particular, "*completeness (or level of completeness) refers to the proportion of landslides shown in the inventory compared to the real (and most of the times unknown) number of landslides in the study area*". Full completeness is
unachievable (Corominas et al., 2014) while "*a substantially complete inventory must include a substantial fraction of the smallest landslides*" (Malamud et al., 2004). More specifically, the completeness of an inventory has often been evaluated in relation to the size of the landslides (Stark and Hovius, 2001) with the expectation that the ratios between the number of landslides present in different size classes is equal or very similar to the ratios observed when considering the whole population of landslides. Guzzetti et al. (2012) stated that only event inventories can be statistically representative, i.e., they contain a
representative sample of the different landslide size classes, while other types of inventories can't (Malamud et al., 2004). Landslide inventories obtained from remotely sensed images are the most recurrent source of information used in landslide susceptibility studies at regional scale (Reichenbach et al., 2018). However, they can suffer from certain limitations related to the image's spatial resolution, the expertise of the operator (for manual and automatic classification) or the slope orientation and shadowing effects (Brardinoni et al., 2003; Jacobs et al., 2017).
In practice, many works devoted to the landslide susceptibility/hazard zoning were and are still based on information acquired from field surveys or from historical inventories and catalogues derived from heterogeneous information sources (Bera et al., 2019; Hussain et al., 2019; Jacobs et al., 2020; Knevels et al., 2020; Meena et al., 2019; Reichenbach et al., 2018; Rohan and Shelef, 2019; Zhang et al., 2019). Historical or field-based inventories often show an abundant quantity of landslides near urban areas or infrastructure, where damage is more frequent, sites are more accessible, and mitigation plans are elaborated
(Guzzetti et al., 1999, 1994; Ibsen and Brunsden, 1996; Steger et al., 2021; Trigila et al., 2010; Wood et al., 2020). Usually, the accumulation of information along roads is particularly rich, highlighting that landslides and transport networks are intrinsically interconnected in terms of process and impacts (Taylor et al., 2020). The reasons of that close relationship are very complex and not fully unravelled (Brenning et al., 2015; Donnini et al., 2017; Giordan et al., 2018; McAdoo et al., 2018; Meneses et al., 2019; Santangelo et al., 2015; Sidle et al., 2014; Sidle and Ziegler, 2012), raising a need to investigate whether
the major availability of roadside landslide information in inventories is purely causal (roads act as predisposing factors) or also depends on other factors, such as visibility matters.

Size, distance and orientation determine the visibility of an observed object (like a landslide) (Bornaetxea and Marchesini, 2021; Domingo-Santos et al., 2011). In the case of field mapping, surveyors often follow roads and observe different portions of the territory from different observation points. Small landslides can be easily detectable if they are close to the path, but not
when they are located far away. In addition, due to their position/orientation, it is possible that even large landslides cannot be detected. In contrast, in a satellite or aerial image, the visibility of the territory is referred to the position of the sensor, and it can be assumed to be almost constant and homogeneous, even though spatial resolution and geometric acquisition may limit the minimum size of landslides that can be detected (Mondini et al., 2014). Therefore, the quality and completeness of an inventory can be intrinsically linked to the data acquisition method.



Some authors have suggested ways to assess the quality and/or the completeness of an inventory. Malamud et al. (2004) proposed the landslides area statistical distribution (Frequency-Area Distribution - FAD) as an indicator of completeness, which was used by many authors (e.g., Chaparro-Cordón et al., 2020; Ghorbanzadeh et al., 2019; Tanyaş et al., 2019; Zhang et al., 2019). According to Malamud et al. (2004), the inverse gamma distribution can be used to model the frequency density of landslide sizes in an inventory. A number of authors consider an inventory statistically representative only for landslides

larger than the roll-over value (Nicu et al., 2021; Roberts et al., 2021; Tanyaş and Lombardo, 2020; Tekin, 2021; Ubaidulloev et al., 2021). Galli et al. (2008) suggested a framework, based on pairwise comparisons (geographical abundance, cartographic matching, frequency area statistics, and effectiveness in modelling landslides), to rank inventories prepared in the same study area. Piacentini et al. (2018) analysed the spatial accuracy of a historical geospatial landslide database comparing different periods within the time laps covered by the catalogue. They also verified the completeness of the database by the conventional

FAD analysis. Trigila et al. (2010) used landslide densities in urban and non-urbanized areas to rank landslide inventories quality across the different administrative regions of Italy. Setting a buffer of 750 m around the urban areas they ranked higher those inventories where the percentage of landslides mapped outside the buffer areas was larger. The approach requires the choice of a fixed buffer a-priori not connected to the local morphology of the area, to the related geomorphological processes and to the visibility of the slopes from urban centres. Finally, Tanyaş and Lombardo (2020) proposed a completeness index

for earthquake-induced landslide inventories. The index is a function of the Peak Ground Acceleration values and therefore cannot be applied to rain-induced landslide inventories or historical archives.

In this work we analyse the possible relationship between the degree of visibility of the territory from roads and the spatial distribution of information on landslides contained in 4 inventories, prepared with different data and methods. In other words, the "data collection effect" (sensu Steger et al. (2021)), on different types of inventories, is investigated and discussed.

**3 Methods**

In this paper we estimated the level of visibility of the territory using roads as observation points for the inventories obtained through field investigation and we assumed constant visibility when remotely sensed images were used.

For field-based maps, we used the solid angle (SA) metric as a measure of the visibility of an object. SA value measures the portion of the observer field of view occupied by the object.

We calculated SA maps using r.survey (Bornaetxea and Marchesini, 2021). r.survey is an open source spatial analysis tool useful to assess how the terrain morphology is perceived by an observer located at a defined observation point, or a group of points. It was designed for evaluating the visibility of features lying on the terrain slopes, including landslides. The positions of the observer, a DTM and the size of the target object whose visibility is going to be assessed (a landslide in this case) are the mandatory inputs. Among the different outputs, the tool provides the map of the maximum solid angle (SA). In the

maximum solid angle map, each pixel has only one value. However, each pixel is potentially observed from several observation points. Here, the pixel value represents the maximum solid angle value calculated among all observation points from which





the pixel is visible. SA value depends on the size of the observed object, the distance (between observers and target) and the relative orientation of the target with respect to the observation point.

The data required to obtain SA are a digital terrain model, a landslide inventory, and road map.  First, we generated a set of

closely spaced points along the roads. These points redundantly simulate the observation points of a surveyor moving along the roads. Then we used r.survey to calculate the maximum SA map for a circular object similar in size to the smallest landslide in the inventory. After that, we obtained the visibility class map (SAc) by thresholding the SA values. Additionally, we smoothed the SAc maps, replacing the central pixel values with the most frequent class (mode) in a 3x3 moving window, in order to remove isolated pixels belonging to different classes with respect to the surrounding ones. Finally, we estimated the

landslide density counting the number of landslides in each visibility class. Since landslides are commonly collected as polygonal areas, it may happen that a single landslide overlaps more than one visibility class. In this case, we assigned the landslide to the most present class within the landslide polygon

We used two metrics to measure the spatial density: the Normalized Landslide Count (NLC) and the Standardized Landslide Density (SLD).

We used NLC to compare the spatial density of landslides included in different inventories prepared for the same study area (Eq. 1):

$$NLC_i = \frac{n_i}{n_t},$$ (1)

where $n_i$ and $n_t$ represent the number of landslides in class $i$ and the total number of landslides, in the inventory, respectively.

We used SLD to compare the spatial density of landslides included in different inventories prepared for different study areas

(Eq. 2):

$$SLD_i = \frac{NLC_i}{(A_i/A_t)},$$ (2)

where $A_i$ and $A_t$ are respectively the area of visibility class $i$ and the total area.

The SLD metric normalises NLC according to the percentage of territory occupied by the visibility classes. These percentages, in fact, can be slightly different among the study areas, due to the smoothing performed to remove isolated pixels.

The entire flowchart is described in Fig. 1.



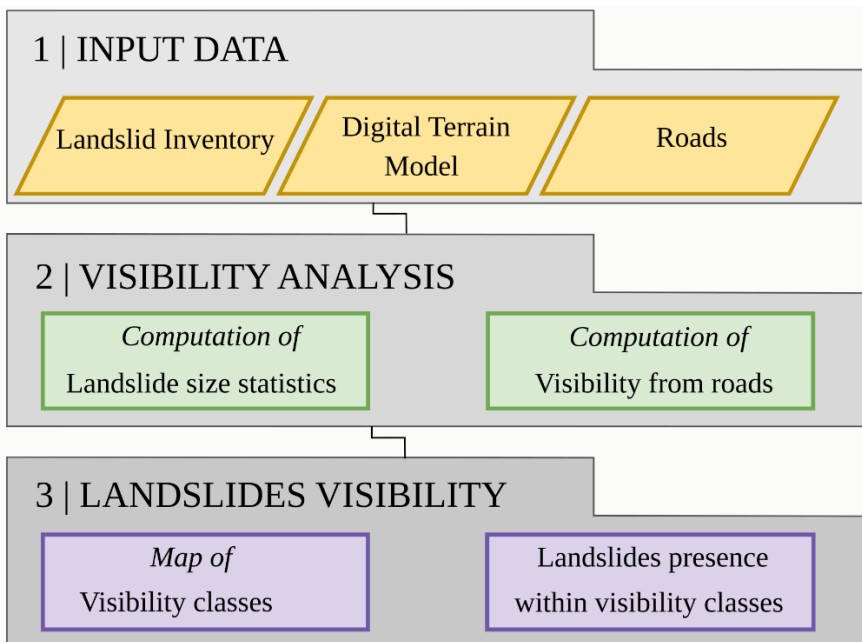

**Figure 1: Conceptual-chart illustrating the proposed GIS-based approach.**

## 4 Study areas

We first tested the approach in Gipuzkoa Province. It is a ~1980 km² region located in the north of the Iberian Peninsula, along
the western end of the Pyrenees (Fig. 2a). It is lithologically heterogeneous, with deposits dated from Paleozoic to Quaternary,
and presents the typical hilly and mountainous Atlantic landscape. The average annual precipitation is 1597 mm, with two
maximum seasons: November–January and April. More detailed description about this study area can be found in (Bornaetxea
et al., 2018).

We applied the same approach in an area of ~513 km² within the Darjeeling district, the northernmost district of West Bengal
state (north-east of India) (Fig. 2b). The area starts just above the foothills of Himalaya in the south and goes beyond the Higher
Himalayas in the north. The area lies within the highly dissected hill ranges of the sub to higher Himalayas with elevation
varying from 200 m to 2900 m. About 48% of the area has slopes between 15° and 30°, however the steeper slopes are mainly
restricted in the escarpment or cliffs present in the area. The major part of the area is covered by Tea plantation (39%), followed
by Moderate vegetation (24%), Sparse vegetation (19%), Thick vegetation (8%), Settlement and Cultivated land (4% each).
The area is a part of active fold thrust belt of Darjeeling Himalayas where sedimentary rocks of Sub-Himalayas, low grade
meta-sedimentaries of lesser Himalayas and high-grade rocks of Higher Himalayas are present with or without the overburden
cover of varied thickness. These sequences of different grades of rocks are separated by E-W trending major tectonic features
like Himalayan Frontal Thrust (HFT), Main Boundary thrust (MBT) and its splay as well as Main Central Thrust (MCT). The
area is located within the seismic Zone-IV of seismic zonation map of India (BIS 2002).



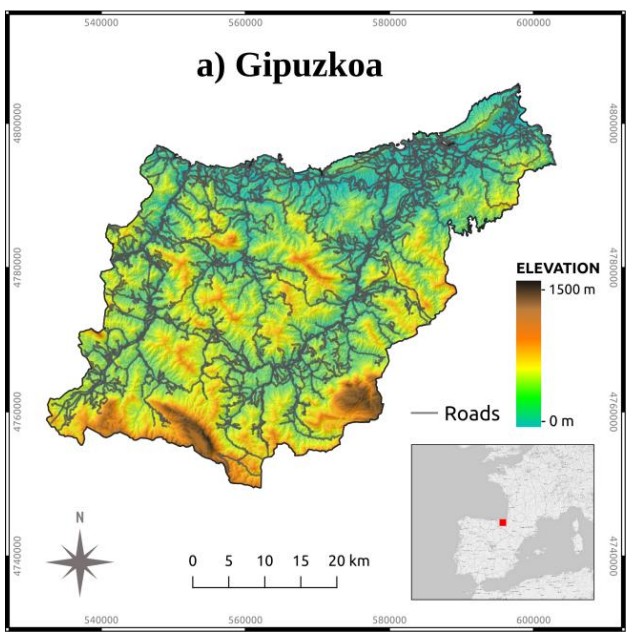 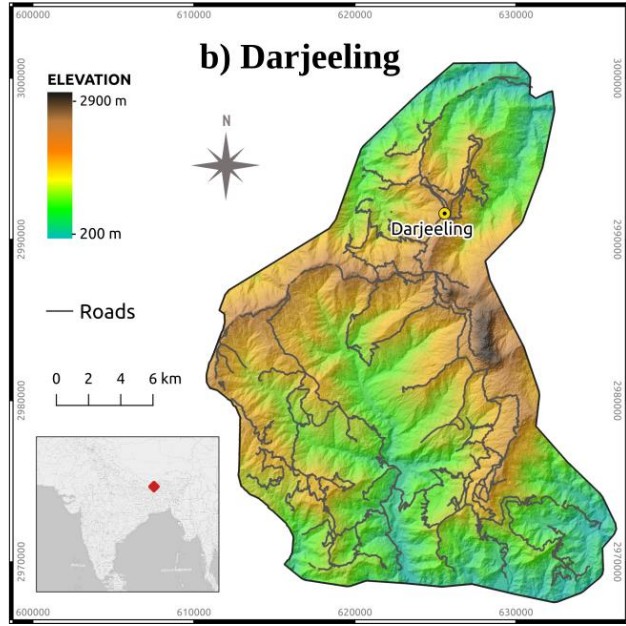


**Figure 2: Location maps of Darjeeling district (India) - Projection: WGS 84 / UTM zone 45N - and Gipuzkoa province (Spain) - Projection: ETRS89 / UTM zone 30N. Location Base Maps: © OpenStreetMap contributors 2021. Distributed under the Open Data Commons Open Database License (ODbL) v1.0.**

The Darjeeling study area experiences a temperate climate with wet summers which gradually moves into monsoon season
when the area receives a number of wet spells, notorious for triggering landslides. This part of the Eastern Himalayas receives the maximum amount of precipitation within the entire Himalayas.

The Darjeeling Himalayas is perennially landslide-prone and frequently experiences landsliding events of variable magnitudes. Most of these landslides are triggered by incessant monsoon rain between June and September, with some occasional major landsliding events in between.

Study areas road networks are shown in Fig. 2. In Gipuzkoa roads are mostly located along the valleys while in Darjeeling roads are usually positioned along the relief's ridges.

## 5 Data

For Gipuzkoa Province, we used a landslide inventory prepared, by one of the authors, during the field-work campaign carried out in the period from June to August 2016 (Bornaetxea et al., 2018). We cleaned the database from landslides not detected by
means of visual inspection on the field, and 542 shallow landslides remained. This inventory is referred to as **Gipuzkoa** inventory. Additionally, the map of the roads followed during the field-work campaign and 5 meters resolution DTM was available. For the Darjeeling study area, the Geological Survey of India (GSI) provided us with an inventory that was the result of a field-work campaign carried out after the monsoon period (that goes from June to September) of 2019. This inventory, named **GSI Field**, provides landslide locations as points, so the FAD curve cannot be computed. Additionally, GSI also





provided us with a historical landslide inventory for the Darjeeling area. It is a multi-temporal landslide inventory devoted to landslide susceptibility modelling and studying triggering mechanisms, landslide domains and mitigation actions. This database gathers information about landslides that have occurred since 1968. As it usually occurs with national or regional multi-temporal databases (Van Den Eeckhaut and Hervás, 2012), the information in this data-base is heterogeneous. Out of 1240 landslides, 80% are represented as polygons, while 20% are single points. Almost half of the landslides (47.6%) were

mapped by means of satellite image photo-interpretation, using the available images coming from diverse sources, such as Cartosat PAN (2%), LISS IV (1%) and Google Earth (44.6%). The rest of the data came mainly from legacy data, including data collected from GSI reports, and Toposheet (34.6%). The latter corresponds to a Topobase map of Survey of India (SOI) surveyed in 1969-70 at 1:25000 scale. Other sources such as Darjeeling Himalayan Railway's database (7.5%), Blogs or Newspapers (3.5%) and Field-work (6.8%) complete the available information. Debris slides (69.43%) and rock slides (18.4%)

are the most frequently reported failures together with debris flows (5.3%), rock fall (0.23%), deep rotational slides (1.95%) and unknown (4.69%). We named this inventory as **GSI Historic**. We obtained the corresponding FAD curve only using the landslides mapped as polygons. The curve (Fig. 3) shows the conventional power-law fit with a relatively low roll-over (Malamud et al., 2004). Lastly, we mapped landslides triggered by the 2019-2020 monsoon season using a pre-event pan sharpened Spot 6 image acquired on 22th March of 2019 and a pan sharpened post-event image acquired on 3th April of 2020

by the same satellite. We used the two 2.5 x 2.5 m spatial resolution images to detect landslides occurring in between the two acquisitions following a photo interpretation approach. In this inventory, referred to as **Spot 2020**, we classified most of the landslides (95%) as earth and debris flows, and the rest as complex movements. Figure 3 shows the power law shape of the FAD curve on the right of the rollover and a low rollover value (Malamud et al., 2004). Table 1 summarizes the information about the four inventories.

In addition to the landslide information, GSI also provided us with the road network map of Darjeeling, together with the 10x10 meters resolution DTM.



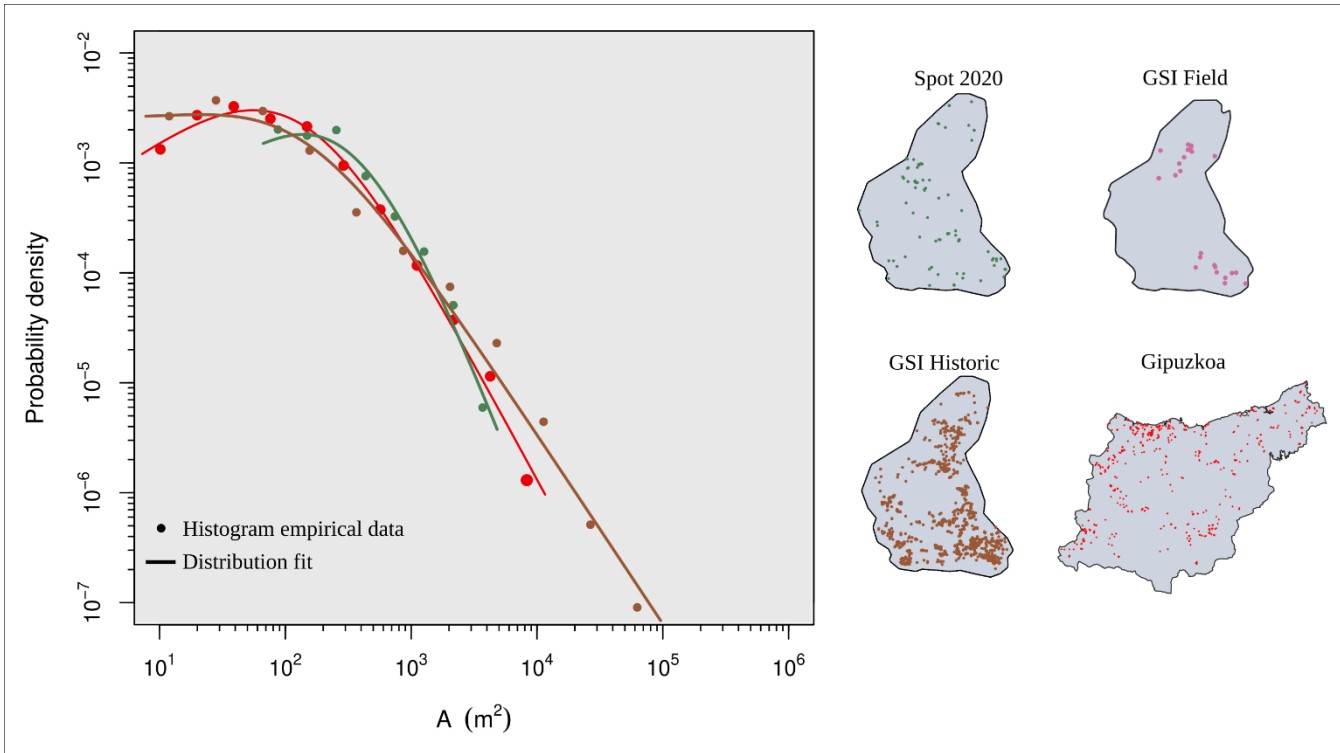

**Figure 3: Frequency area distribution curves (FAD curves) for Spot 2020 (green), GSI Historic (brown) and Gipuzkoa inventories (red). Landslide distribution map for Spot 2020, GSI Historic, GSI Field and Gipuzkoa inventories.**

|  | **Gipuzkoa** | **Spot 2020** | **GSI Historic** | **GSI Field** |
|---|---|---|---|---|
| **Number of landslides** | 542 | 82 | 1240 | 25 |
| **Source** | Field survey | Satellite image photo-interpretation | Miscellaneous | Field survey |
| **Geometry type** | Polygons | Polygons | Points and polygons | Points |

**Table 1: Descriptive table for the landslide inventories.**

## 6 Visibility class maps

We obtained the visibility class maps for field-based inventories in the two study areas using the settings listed in Tab. 2. In Gipuzkoa we deployed points every 200 m along the road paths followed during the field-work. According to the experiments carried out in Bornaetxea et al. (2018) 200 m was considered suitable for this concrete case. In Darjeeling we

knew that field-based inventories were produced by visual inspection from roads but not the real path followed by the surveyors. As a consequence, we used the entire road network (including roads slightly outside the boundaries of the studied area) and a maximum distance between points of 50 m for modelling the visibility.



We set to infinity (Tab. 2) the maximum line of sight distance in order to assess the visibility level for the complete territory.

We calculated maximum solid angle (SA) maps for hypothetical landslides with an area similar to the smallest landslides

inventoried in Gipuzkoa and Darjeeling (Tab. 2), and we assumed that larger landslides were therefore visible.

|  | **Gipuzkoa** | **Darjeeling** |
|---|---|---|
| **Distance between points** | 200 | 50 |
| **Number of points** | 14352 | 11054 |
| **Maximum visible distance** | infinity | infinity |
| **DTM resolution (m)** | 5 | 10 |
| **Target Object size (m²)** | 19.63 | 78.54 |

**Table 2: Summary of the specific settings to calculate *SA* maps for each study area.**

We classified the *SA* maps in 6 classes (*SAc map*), using 16.67th, 33.33th, 50th, 66.67th, and 83.33th and 100th quantiles as thresholds. Then we applied the 3x3 moving window smoothing. Details about the *SAc* maps and the threshold values for each class are available in Tab. 3 and Fig. 4.

|  |  | **Gipuzkoa** |  | **Darjeeling** |  |
|---|---|---|---|---|---|
| Quant. | Class | Bin (min²) | Area (km²) | Bin (min²) | Area (km²) |
| 100.0 | 1 | 4897.85 – 74141600 | 334.93 | 4561.02 - 74141601 | 90.87 |
| 83.33 | 2 | 1039.20 – 4897.85 | 329.46 | 942.98 - 4561.02 | 88.124 |
| 66.67 | 3 | 271.42 – 1039.20 | 328.31 | 345.36 - 942.98 | 85.29 |
| 50.0 | 4 | 63.27 – 271.42 | 328.23 | 150.36 - 345.36 | 83.27 |
| 33.33 | 5 | 4.67 – 63.27 | 328.18 | 59.74 - 150.36 | 81.94 |
| 16.67 | 6 | 0.0 – 4.67 | 326.99 | 0 - 59.74 | 84.21 |
| **Total** |  |  | **1976.12** |  | **513.70** |

**Table 3: Details about the visibility classes for Gipuzkoa and Darjeeling study areas. The abbreviation Quant. Refers to quantiles. min² stands for square minutes, a unit of measure of the solid angle.**

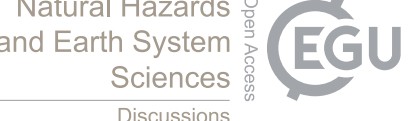

**Figure 4: Visibility class map of Gipuzkoa and Darjeeling study areas. Red lines in the zoom insets represent the roads used as reference observation points.**

The territory in the different visibility classes is quite homogeneous in terms of morphometry, lithology and land use as shown in the plots of (i) the probability density of the Slope values and (ii) the percentual spatial coverage of the Lithology and Land Use categories (Fig. 5).

The slope distribution in Gipuzkoa (Fig. 5a) shows a local maximum for the very low slope values in visibility classes 1, 2 and 3. This is probably due to the fact that roads in Gipuzkoa are mainly on the valley floor, and consequently the plains are

located in the most visible portions of the territory. In Darjeeling (Fig. 5b), where roads are primarily located along ridges, slope values are very homogeneous among the visibility classes. Concerning lithology, in Gipuzkoa (Fig. 5a) only the slate rocks show a relevant difference in class 6. This is due to the localized outcrop of this metamorphic material in the eastern part of the territory. Regarding the land use, although the general trends are always homogeneous, anthropic and grass land uses are dominant in the most visible classes (classes 1 and 2), while forests and scrubs and hedges are more abundant in the

less visible classes. In Darjeeling (Fig. 5b), lithological and land use categories are also similarly represented within the different visibility classes.

**Figure 5: Slope probability density plots and Land Use and Lithology distribution by visibility classes for Gipuzkoa and Darjeeling study areas. In Gipuzkoa land use types are Agr: Agricultural; Ant: Antropic; Bea: Beach and peatlands; For: Forest; Grss: Grass; Rck: Rock; Scr: Scrub and hedges; Wtr: Water. The Lithology types are ClDt: Clay and Detrital rock; Lms: Limestones; Mgm: Magmatic rocks; Mar: Marls; No: No rock; Slt: Slate; Srd: Surface deposits. In Darjeeling land use types are Br: Barren; Cult: Cultivated land; Mveg: Moderate vegetation; Riv: River; Stl: Settlement; Spr: Sparse vegetation; Tea: Tea plantation; Tveg: Thick vegetation; Wt: Waterbody. The lithology types are Mig: Banded migmatite, Gt-Bt gneiss, mica schist, biotite gneiss; Brw: Brownish, yellow oxidised soil with boulders-pebbles and latsol; Cgn: Calc granulite, quartzite, gneiss, Gar, Sil, Kya schists; Csch: Chlorite**






**sericite schist and quartzite, meta-graywacke; Myl: Mylonitic granite gneiss; Qrz: Quartz arenite, black slate, cherty phyllite, quartzite; Snd1: Sand, silt and clay; Snd2: Sandstone, clay, shale, conglomerate; Snd3: Sandstone, shale with minor coal.**

## 7 Description and analysis of the results

In Fig. 6 we show NLC values for the Gipuzkoa Inventory, which is a strictly field based landslide database. Figure 6 highlights that the majority of landslides are located in very visible areas i.e., classes 1 to 3, and only a negligible number of

landslides is located in scarcely visible areas (class 5 or 6). Data show a near monotonic decrease of the NLC as the level of the visibility decreases, or what is equivalent, as the visibility class increases.

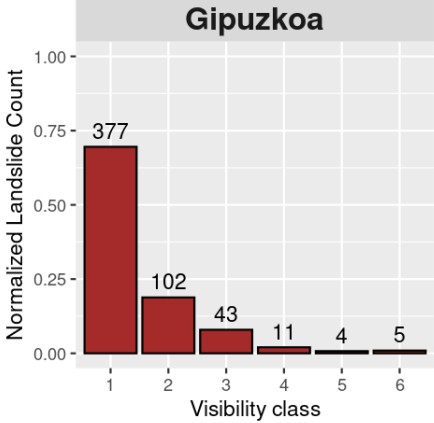

**Figure 6: Normalized landslide count distribution plot for Gipuzkoa inventory. The values above each column indicate the number of landslides in each visibility class.**

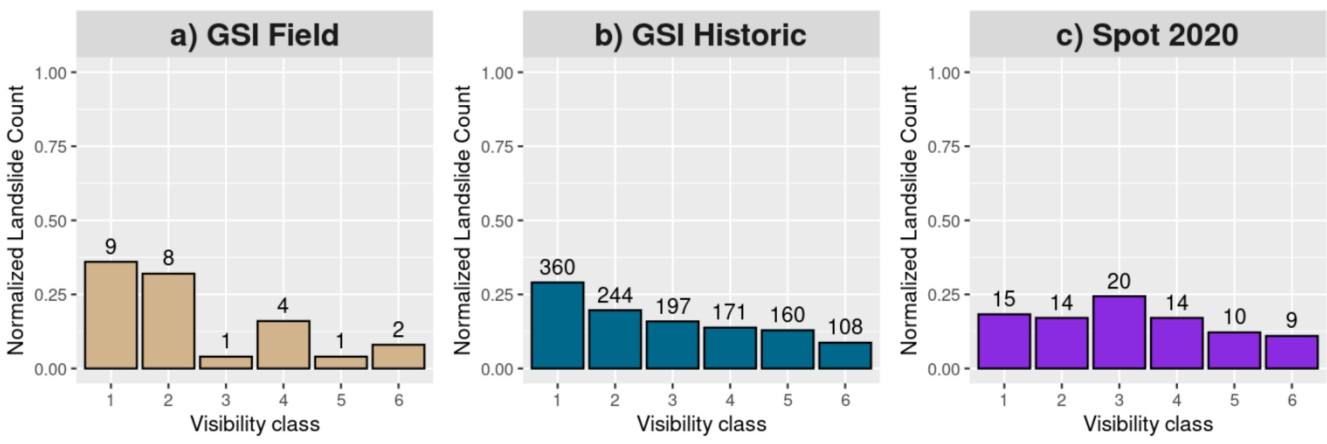


**Figure 7: Normalized landslide count distribution plot for GSI Field, GSI Historic and Spot 2020 inventories. The values above each column signify the number of landslides in each visibility class.**

Figure 7 shows NLC versus visibility classes for the landslide inventories of Darjeeling. In GSI Field (a field-based inventory), we used single points to locate the (few) landslides (Fig. 7a), and we sampled visibility class values from the





pixel in which they fell. As expected, most of the landslides are located within the most visible classes (class 1 and class 2).
The values in the other classes are very fluctuating but this is probably due to the fact that the number of landslides in the
inventory is very small.

GSI Historic (Fig. 7b) includes landslides mapped using different methods and shows a slight, but still monotonic,
decreasing trend.

Figure 7c shows the calculated NLS values for the Spot 2020 inventory, produced solely by photo-interpretation of satellite
imagery. The values calculated in the visibility classes are fairly homogeneous and show a non-monotonic trend.

GSI Historic inventory contains two main types of information: landslides mapped exploiting satellite/aerial images or
collected during field-based survey and from legacy data. We separated data obtained by satellite/aerial images from the rest
of the data sources and called them GSI Historic Sat and GSI Historic Others respectively.

NLC values show a pronounced monotonic decreasing trend for the GSI Historic Others inventory (Fig. 8a), copying the
pattern observed for the field-based Gipuzkoa inventory (Fig. 6).

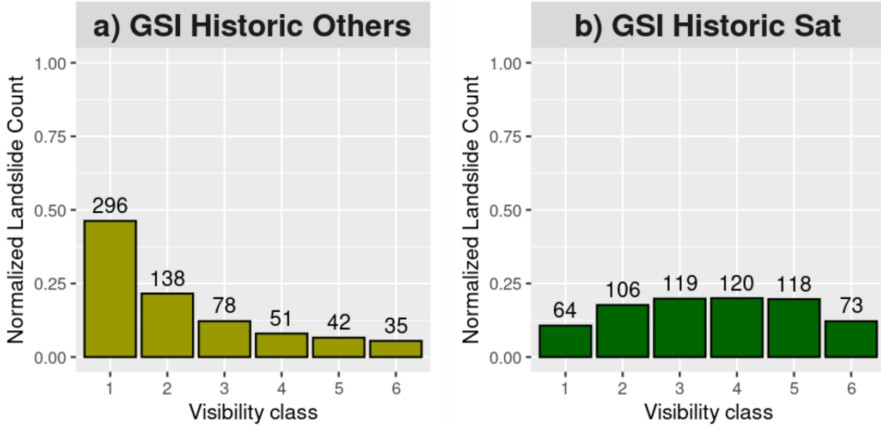

**Figure 8: Normalized landslide count distribution plot for GSI Historic Others and GSI Historic Sat inventories. The values above
each column signify the absolute number of landslides in each visibility class.**

The GSI Historic Sat (Fig. 8b) shows a similar trend to that observed for the Spot 2020 inventory (Fig. 7c), with the
landslide density not dependent on visibility classes.

We compared landslides sizes in the different visibility classes (Fig. 9). In Gipuzkoa and GSI Historic Others we merged
respectively classes 3,4,5,6 and 4,5,6 to have enough samples. We did not consider Spot 2020 and GSI Field inventories
because of the little amount of data.
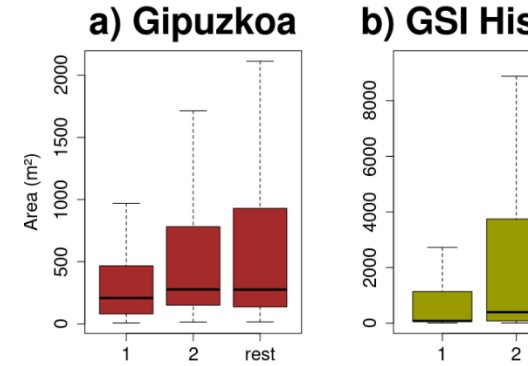
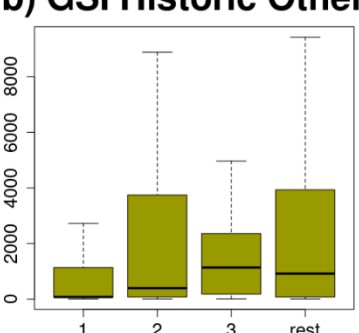
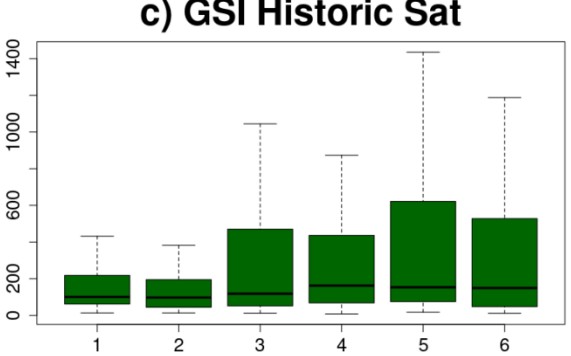


**Figure 9: Landslide area boxplots per visibility classes. Numbers in abscises are the visibility classes.**

Landslides in class 1 are significantly smaller than those included in the other classes, for both Gipuzkoa and GSI Historic Others inventories (Fig. 9a and 9b). Furthermore, the landslide size central value in these two inventories tends to increase with the decrease of the visibility class, while the maximum variation of the median is 188% and 2000% respectively. For

the GSI Historic Sat landslide sizes (Fig. 9c) we did not observe a clear increasing trend in the median value, which shows a considerably smaller maximum variation, 66%.

We also compared the standardized landslide density (SLD) values with respect to the central value of each visibility class (Fig. 10), in order to allow a comparable contrast between all the inventories (visibility values were plotted in logarithmic scale). We excluded GSI Field and GSI Historic inventories from this analysis due to data scarcity or because of the non-

homogeneous source of the data. In Fig. 10 Gipuzkoa and GSI Historic Others show a monotonically non decreasing behaviour where the amount of landslide increases with the solid angle value, i.e., when the terrain visibility from the roads increases. In contrast, GSI Historic Sat and Spot 2020 show an almost flat behaviour, where the number of landslides does not vary according to level of visibility from the roads.

We further simulated the visibility of landslides included in the GSI Historic Sat and Spot 2020 inventories from roads. By

setting a visibility threshold of 400 square minutes (which is slightly larger than a person's maximum visual acuity (Bornaetxea and Marchesini, 2021; Healey and Sawant, 2012), we calculated the SLD values for the landslides potentially visible from roads (Fig. 10). In this scenario, the number of (potentially) visible landslides is 55 (-32.9%) and 301 (-59.8%) for **Spot 2020 visible** and **GSI Historic Sat visible** respectively. Furthermore, for values of log(SA) smaller than about 3.5 and 3.0, the graphs of GSI Historic Sat Visible and Spot 2020 Visible respectively show a monotonically increasing pattern.

In the simulation, missing landslides are mostly in areas with poor visibility.





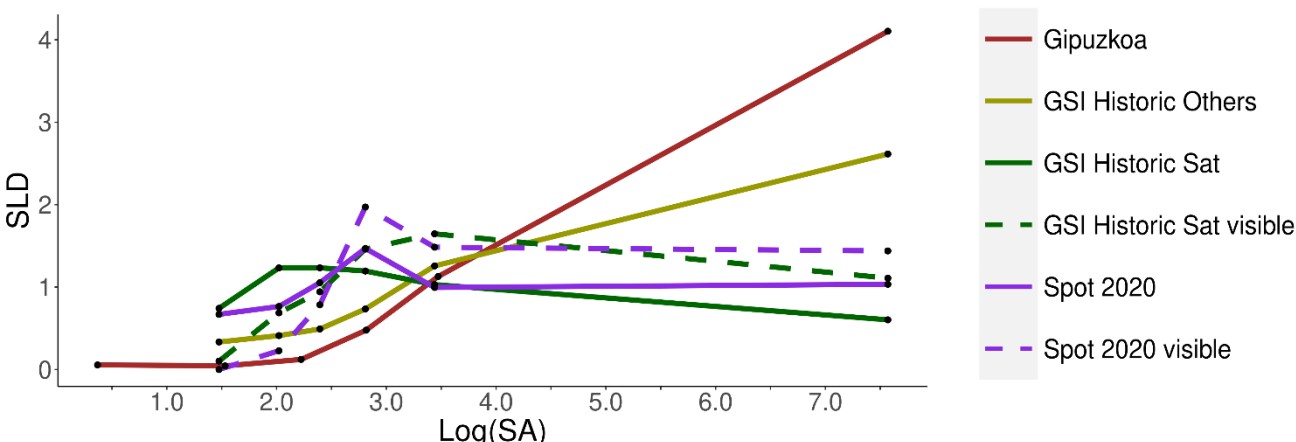

**Figure 10: Normalized landslide count distribution plot for GSI Historic Others and GSI Historic Sat inventories. The values above each column signify the absolute number of landslides in each visibility class.**

**8 Discussion**

We analysed the relationship between the terrain visibility and the information on landslides in four inventories prepared with different data and methods. In particular, we contrasted information on the density of landslides collected in the field or by archive data collection with those obtained through photo interpretation of satellite images. For surveys conducted from roads we modelled visibility using a geometric approach taking into account the local morphology of the territory. The most visible areas are generally located in the vicinity of roads, with a progression often differing substantially from geometric

buffering. This is shown in Fig. 4a (Gipuzkoa) and in Fig. 4b (Darjeeling) by the lack of symmetry of the boundaries of the classes (with respect to roads) and by the presence of portions of territories very close to roads but not visible from roads. In the field based Gipuzkoa inventory, landslide density correlates positively with terrain visibility, showing a monotonic decreasing trend (Fig. 6). Furthermore, the median size of the landslides is larger in the less visible classes (Fig. 9a). The spatial densities of landslide information and visibility classes are positively correlated also in the GSI Field (Fig. 7b)

and GSI Historic Others (Fig. 8a) inventories. The deviations from a monotonic trend, observed in the GSI Field inventory, are probably related to the low data density and location inaccuracy. We assert that limited visibility or complete lack of visibility from observation points along the roads hampered the possibility to detect landslides, in particular when small. In the inventories based on satellite imagery (Spot 2020 and GSI Historic Others) the density of landslides in the visibility

(from roads) classes is quite uniform (Fig. 7c and 8b). Furthermore, for the GSI Historic Sat inventory, the variation of landslide size in the different visibility classes does not show a clear trend and presents much smaller variations than the Gipuzkoa and GSI Historic Others inventories. This is assumed to be a consequence of the neutral observation point offered by the remote acquisition.





Visibility can explain the lower capacity of field surveyors to detect landslides located in remote areas, and the higher

capacity to detect small landslides along the road. On the other hand, landslide mapping through satellite image photo

interpretation showed an overall homogeneous performance, mainly because the single observation point offered a constant

visibility level along the territory. In this case, limits in the spatial resolution and in the acquisition geometry, can be the

factors hampering the possibility of mapping small landslides (Mondini et al., 2014).

The areas covered by each visibility class are similar in terms of landslide predisposing factors such as morphometry,

lithology and land use (Fig. 5). Roads, present only in the first visibility class, are also considered by many authors as a

predisposing factor since cut-and-fill failures, drainage and groundwater alteration can influence the occurrence of landslides

(Brenning et al., 2015; Donnini et al., 2017; Giordan et al., 2018; McAdoo et al., 2018; Meneses et al., 2019; Santangelo et

al., 2015; Sidle et al., 2014; Sidle and Ziegler, 2012). Based on the above considerations, a higher density of landslides

should be expected in class 1 than in classes 2 to 6, with the latter characterised by fairly similar density values. This was not

observed in inventories based on satellite imagery nor in those acquired in the field. In both types of inventories, there is a

data collection effect, albeit different.

In fact, since roadside landslides are typically small (Voumard et al., 2018), they can be easily under-represented when the

inventories are prepared with images without an adequate resolution (Martha et al., 2021). On the contrary, when

considering inventories based on field surveys and historical data collection (Gipuzkoa, GSI Field and GSI Historic Others),

small landslides are very abundant in the highly visible areas, and very few landslides are intercepted in the low visible

areas. This suggests that the visibility can affect the spatial (Fig. 6, 7a,b) and the size (Fig. 9) distributions of the reported

information. These results are in line with Steger et al (2021) hypothesis on the "data collection effect", which assumes that

the method used to compile inventories can influence the spatial distribution of landslides information in the inventories.

We observed a partial monotonic behaviour of the relationship between visibility and landslide density obtained in the field-

based inventories also in Spot 2020 Visible and GSI Historic Sat Visible. These two inventories include only those

landslides present in the remotely sensed inventories that would result visible through hypothetical field surveys along the

roads. SLD values (Fig. 10) highlight that the least visible areas (low SA values) lost the majority of landslides. The

simulation confirms that the abundance of landslides observed in the most visible areas (high SA values), in inventories

prepared by field survey and/or legacy data, was not achieved by the remote sensing-based inventory. Indeed, in the most

visible areas, SLD values from the Spot 2020 Visible and GSI Historic Sat Visible inventories are consistently much lower

than those observed for the GSI Historic Others and Gipuzkoa inventories. This suggests a possible lack of small landslides

detected in visibility class 1 for Spot 2020 and GSI Historic Sat caused by the inadequate resolution of the images. Thus,

also inventories prepared using inadequate images may be affected by the "data collection effect".

Our results depend on some user-driven decisions, such as the distance between observation points placed along the roads,

and the choice of the thresholds to obtain the visibility classes. In Darjeeling, the real path followed by the field surveyor is

unknown and we applied a conservative approach (visibility overestimation) by considering all roads as potential

observation points and a four-time smaller maximum distance between points than in Gipuzkoa. Furthermore, we chose



quantile values of the SA map to obtain visibility classes covering similar portions of the study area. We run several tests with different threshold values showing results in terms of landslide densities always very similar to those described by Figs.

6, 7, 8, 9, and 10.

The resolution of the DTM can also have an impact on the results. A coarse representation of the morphology of the territory can affect the calculation of the solid angle and the delineation of non-visible areas. In this work, we performed the analysis with the higher resolution DTMs available in each study area, but, further tests on the influence of the quality of the data should be conducted. Future works should also incorporate the role of the vegetation in the visibility for landslide detection

by field-work, although the information about the elevation of each type of vegetation is rare.

We think that the procedure and methods presented in this work can be used to: (i) test whether the spatial distribution of landslides in existing inventories (especially those created by fieldwork) is affected by a marked correlation with visibility from observation points, (ii) identify portions of land where landslide density information is enough accurate to calibrate susceptibility, hazard and risk models with more robust data, (iii) identify portions of land where landslide inventories need

improvement, (iv) plan exhaustive field mapping campaigns.

**Conclusions**

We analysed the relationship between the spatial density of landslides reported in different inventories prepared through field surveys, collection of previous data and interpretation of remotely acquired images, and the visibility of the territory from observation points located along the roads.

The results of the present work show that in inventories prepared using field survey and/or historic legacy data, the spatial distribution of landslides can be strongly affected by the "data collection effect". This is demonstrated by (i) the positive correlation observed between landslide density and the visibility of the terrain from the observation points, (ii) the lack of small landslides in areas with low visibility, and (iii) a number of landslides in remote areas intercepted by the images but invisible from roads.

In addition, inventories based on the use of remote sensing images, can also be affected by a "data collection effect". In fact, results show that, contrary to what expected (Brenning et al., 2015; Donnini et al., 2017; Giordan et al., 2018; McAdoo et al., 2018; Meneses et al., 2019; Santangelo et al., 2015; Sidle et al., 2014; Sidle and Ziegler, 2012), our inventories don't show abundance of landslides close to roads. Reasons may be searched in the inadequate spatial resolution of the satellite images, that can prevent the recognition of small roadsides landslides.

Thus, our inventories proved not to be uniformly representative of the real spatial distribution of landslides in the study area, requiring for an informed and appropriate usage (Bornaetxea et al., 2018; Steger et al., 2021). Our procedure enriches the portfolio of solutions to evaluate the quality of landslide inventories introducing local morphology in the analysis.
**Data availability**

The digital elevation model of Gipuzkoa Province with $5 \times 5$ meters of resolution, was downloaded from the official geospatial
data repository www.geo.euskadi.eus. The Gipuzkoa inventory and the field work paths are data generated by the authors and
are available under request. The digital elevation model of Darjeeling with $10 \times 10$ meters resolution, roads map of Darjeeling
and all the landslide inventories used in this work were provided by the Geological Survey of India. The tool r.survey is
available in https://doi.org/10.5281/zenodo.3993140.

**Author contribution**

TB and IM conceptualized the work and designed the overall methodology. TB carried out the analysis with the technical
assistance of IM and AM. SK and RK provided the necessary data and contributed to interpret the results through their local
perspective. TB prepared the first version of the manuscript and IM and AM contributed enormously to its revision and edition.
AM coordinated the work and obtained the necessary funds.

**Competing interests**

The authors declare that they have no conflict of interest.

**Acknowledgments**

The work was partially funded by the UKRI Natural Environment Research Council's and UK Government's Department for
International Development's Science for Humanitarian Emergencies and Resilience research programme (grant number
NERC/DFID NE/P000649/1). Txomin Bornaetxea was financially supported by the postdoctoral fellowship program of the
Basque Government (grant numbers POS_2020_2_0010) in the framework of a scientific collaboration with the Geological
Survey of Canada and during the scientific collaborations with the Geomorphological Group of the Research Institute for the
Geo-Hydrological Protection in Perugia, Italian National Research Council (CNR-IRPI).

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
