# Peer review of "Terrain visibility impact on the preparation of landslide inventories: some practical cases"

_Natural Hazards and Earth System Sciences, 2021_

## Author Response (AR1)

**Manuscript nhess-2021-334**

We thank the reviewers for their useful suggestions.

We have carefully considered their comments and significantly revised the paper. We completely restructured the manuscript following the reviewers' recommendation and this implied some mayor changes: (i) Introduction and Discussion sections have been completely reformulated; (ii) the order and names of the previous Methods, Study Areas, Data, Visibility Class Maps, Descriptions and Analysis of Results sections were modified and; (iii) Results of the experiment in Gipuzkoa study area were moved to the new specific section 4.3.

Below we list the reviewers' comments, our responses, and, where possible/needed, we explain how the text was amended.

**Replies to the comments of #Referee 1**

(num) *comments from referees*, (●) *author's response*, (-) *author's changes in the manuscript*

1) **Maybe it would be more correct to compare different methods of landslide inventory for the same area. In other words, you probably need the same number and types of inventories for each of the study regions to compare.**

   - The suggestion of referee #1 is correct, the asymmetry is clear but unfortunately there are no satellite-based landslide inventories available for Gipuzkoa that allowed us to carry out exactly the same comparison done in Darjeeling. We limit now the use of the field-based Gipuzkoa inventory to show that the monotonically descending trend of landslides density is something that can emerge from inventories obtained by these approaches.

   - We modified the manuscript in the abstract and in the introduction, in order to remove the references to the Gipuzkoa study area. We also moved all the results related to Gipuzkoa Province to the new section 4.3, in order to maintain the focus in the proposed framework and its application in Darjeeling. Then we address to the figures obtained in Gipuzkoa just as a signal of the confirmation of the behaviour we observed in Darjeeling.

2) **The structure of the manuscript is complicated; separate sections contain repeating information. I recommend reorganizing the manuscript as follows: Introduction; Study area; Materials and Methods; Results; Discussion; Conclusion.**

   - We appreciate the valuable suggestion of the referee # 1 and modified the organization of the sections as suggested.

   - The manuscript is now organized as Introduction; Study area; Methods and Data; Results; Discussion and Conclusions. Where, Methods and Data section is subdivided in Methods and Data and Results section is divided in Classified Estimated Visibility map, Description and analysis of NLC plots and Testing the method with external and modelled data.

3) **Sec. 2 Rationale is a part of the literature review and should be combined with the Introduction; repeated sentences in both sections such as, for example, lines 28-30 and 54-57, lines 25 and 64-65, lines 37-38 and 112-114 should be merged / removed.**

   - We acknowledge the presence of unnecessary redundancies. Therefore, we will follow the suggestions of referee #1 and merge the two sections in one.

   - The manuscript has now only one section called Introduction, which compresses the information provided in the former Introduction and Rationale sections in a more efficient way.

**4) Sec. 3, 5 and 6 should be combined in one Section Materials and Methods (you can divide it into subsections if necessary) and reduced by removing repeated phrases. All the results obtained in the study should be collected in the Results Section and discussed in the Discussion. You can also combine Results with Discussion.**

- As stated in the previous comments, we followed the suggestions of referee #1 and reorganize the manuscript as defined in point 2.

**5) Line 123 – Please decipher the abbreviation DTM at the first mention.**

- We agree.

- We added the meaning of the acronym DTM in the new section 3.1 line 117.

**6) Line 170 Figure 2: In the Figure caption, please start with the description of Fig. 2a, and then of 2b (and not vice versa)**

- We agree.

- We modified the former Figure 2 (current Figure 1) and removed the presence of Gipuzkoa province. In fact, this is a study area previously described in other works that we cite. The analysis concerning this area have been now moved to section 4.3. Figure caption has been modified accordingly.

**7) Fig.2a - On the map, the roads are shown in black, while in the legend they are shown in gray. Please make these designations the same color.**

- This is right. This is probably the effect of the transparency given to the roads layer.

- Former Figure 2 is now Figure 1 and it contains only the map of Darjeeling. We revised the colour of the roads in both legend and on the map in order to be coincident.

**8) Figure 9 is unclear and needs explanations. What is "the landslide size central value"? What is "the median value"? Are these the same terms? What do the dotted lines and horizontal bars mean in the Figure?**

- According to the general conventions in statistics, boxplot diagrams contain horizontal bars that represents the 2nd and 3th quartile limits, a horizontal line that represents the median value and two vertical lines, also called as whiskers, that represent the minimum and maximum values (see the figure on the right). Indeed, in this part of the manuscript we used the term "central value" as a synonym of the median value.

[Figure]

- Former Figure 9 is now Figure 8 and it has been modified. We added additional explanations in the caption and also modified the text by the end of the new section 4.2, in order to avoid confusion between the terms "central value" and "median". Indeed, we removed the term central values (lines 230 - 234).

**9) Line 292 – "Landslides in class 1 are significantly smaller than those included in the other classes" – Why? How can you explain this findings?**

- Visibility class 1 corresponds to the most visible portion of the terrain when observed from roads. The boxplot diagrams in Figure 8 (former Figure 9) represent the landslide areas'

distribution per SA classes. The significantly lower median value in class 1 for GSI Historic Others inventory (and also in Gipuzkoa in the previous version) implies that this inventory contains a considerable amount of small landslides accumulation just in class 1. We further discuss the probable reasons of such differences in the Discussion section (sec. 5 lines 302-312). This is because 1) the approaches followed to collect landslide information in this inventory allow to be more detailed in the most visible area than in the rest of the territory and; 2) there are evidences that 'small' landslides can occur more frequently along the roads because of man-made activities reasons (see references in sec. 5 lines 302-312) than in natural areas. The lack of small landslides in other visibility classes is in our opinion partially due to the data collection approach.

- We added further explanation about this finding in the discussion section (sec. 5 lines 302-312).

**10) Lines 293-294 – "Furthermore, the landslide size central value in these two inventories tends to increase with the decrease of the visibility class, while the maximum variation of the median is 188% and 2000% respectively." I cannot understand this sentence. Please rephrase. Percentage is a fraction of 100. As far as I understand, it cannot be 188 and 2000.**

- The first part of the sentence wanted to explain that, in general terms, landslides in class 1 are smaller than in the rest of the classes for Gipuzkoa and GSI Historic Sat (in Fig. 9 of the former version of the manuscript), and that the lower is the visibility the larger are the mapped landslides. The second part of the sentence just wanted to stress this finding and give some numerical dimensions to the difference between the median in class 1 and the median in classes with less visibility. However, we acknowledge the unnecessary numerical comparisons to explain a rather simple idea. Therefore, the comments of the new Fig. 8 have been modified and rephrased as suggested by the referee #1.

- We modified the description of this finding at the end of the new section 4.2 (lines 230 - 234) and added further interpretations of them in the discussion section.

**11) Figure 10 The caption does not match the picture. "The values above each column signify the absolute number of landslides in each visibility class." - There are no columns in the Figure, no numbers above them.**

- It is right.

- Former Figure 10 is now Figure 9 and we corrected the mistake by adding the right caption.

**12) Lines 334-338 - This is a fairly obvious conclusion, understandable initially and without the use of complex mathematical methods.**

- Referee #1 is partially right. Although it is conceivable that land visibility may have a significant effect on the quality of landslide inventories made by observing land from roads, to the best of our knowledge this effect has not yet been numerically assessed. In order to make it clearer, in the new version of the manuscript we introduced the concept of the "uniformity in the Capacity of Landslides Mapping (CoLM)" (lines 52 - 73). Furthermore, we are not aware that a method based on estimating the visibility of objects of a specific size has ever been proposed to analyse the inventories from this perspective.

- In order to avoid confusion, we eliminated this sentence from the new Discussion section and we reformulated the complete chapter in order to address to the findings obtained by applying the proposed framework. See new section 5.

**13) I believe that the main conclusion from this study may be that the compilation of a landslide inventory requires a combination of both types of methods: remote sensing and field surveys.**

- • We believe that the procedure and methods presented in this work can be used to assess the available inventories and their CoLM uniformity. Our method is able to highlight information gaps along the study area and provides the user of the inventory with valuable information. Then different strategies could be adopted to overcome such information gaps, such as the one proposed by the referee #1.

- - The new section 5 and section 6 synthesize in a more direct way the lessons learned by the experimentation of the proposed method.

**Replies to the comments of #Referee 2**

(num) *comments from referees*, (●) *author's response*, (-) *author's changes in the manuscript*

**1) It is obvious that only visible landslides can be mapped during a field survey, certainly if you do your field survey from a distance. You also explain, and cite, this in your Introduction/Rationale in lines 89-93, even with a comparison to remote sensing products. And you even state you "expected" those results in line 270.**

- • We acknowledge that we did not make clear our assumptions and we created some misunderstanding by overusing the term "visibility". As stated by referee #2, it is conceivable that land visibility may have a significant effect on the quality of landslide inventories made by observing land from roads, but to the best of our knowledge this effect has not yet been numerically assessed. Furthermore, we are not aware that a method based on estimating the visibility of objects of a specific size has ever been proposed to analyse the uniformity in the Capacity of Landslide Mapping (CoLM) in landslide inventories. Our aim was to provide a new method that assess the influence of the visibility constrains in landslide inventories and to test this method with different inventories.

- - We modified the Introduction section and reformulated the objectives of the work (lines 71-77). Also, in order to avoid misunderstandings, we defined the term of Estimated Visibility (EV) as a computer-based simulation of the visibility of an object from given observation point (or points) and; the concept of the Capacity of Landslide Mapping (CoLM) as a signal of the variability in the working conditions that may affect the quality/completeness of the inventory (lines 57 - 77).

**2) As I see it, the present work is basically a validation of your method from a previous work to calculate visibility.**

- • In our previous work we presented a tool for the estimated visibility simulation, that can be used for many different applications. In the current work, we present a framework to explore the uniformity in the Capacity of Landslides Mapping (CoLM) in landslide inventories, which uses r.survey. But to the best of our knowledge, we are not aware that a method based on estimating the visibility of objects of a specific size has ever been proposed to analyse the uniformity in the Capacity of Landslide Mapping (CoLM) in landslide inventories.

- - In order to better explain the novelty of the approach and the general aim we modified the Introduction section and reformulated the objectives of the work (lines 71-77).

**3) On Introduction/Rationale, in your case, I suggest to not separate Introduction and Rationale. The first part of your Rationale is basically a repetition of the introduction, and on the other hand, the introduction lacks information, for example on what you mean by "visibility".**

- We agree. In the new Introduction section some basic concepts are clearly defined, such as the "estimated visibility", which refers to the estimated value simulated by r.survey.

- The new version of the manuscript has now only one section called Introduction, which compresses the information provided in the former Introduction and Rationale sections in a more efficient way.

**4) The question of completeness takes a large part in your Rationale (L62-94), but it is not clear how you want to apply this in your work.**

- The term "completeness" in geomorphology has different interpretations, even considering the definition provided by Guzzetti et al. (2012). The observation of the referee #2 is correct and we have reduced significantly this part, so we just introduced the concept of completeness as it is referred in the literature.

- We reformulated the paragraph between lines 62-94 in the former manuscript and incorporated it into the new Introduction section with more clear and direct statements (lines 40-50 of the new version).

**5) You only state your goal very briefly in L112-114, this needs to be better connected to what you said before.**

- We agree that this idea was not linked well enough to the introduction of the problem.

- With the last two paragraphs in the new Introduction section we tried to clearly state the objective of the study and to link it with the explanations exposed in the previous paragraphs.

**5) It is often not clear to me, what purpose the content of these serve, i.e. , what is the relation of lithology to your research question? You only briefly state that lithology is homogeneous in your visibility classes in your discussion.**

- The purpose of the analysis showed in current Figure 5 corresponds to a test. We wanted to exclude that the unbalanced distribution of landslides in inventories might depend on some lithological/landuse/slope settings in the different SA classes. Since lithology and other landslide predisposing geo-settings are almost homogeneous in the different classes, we are quite confident that the unbalanced distribution may depend on other factors.

- In the new version of the manuscript this is better explained at the end of the new section 4.1 (lines 188-193).

**6) L44-47: This structural information is unnecessary, since it basically just states the headings of the following sections.**

- We accept the suggestion of the referee #2.

- This paragraph has been removed from the new version of the manuscript.

**7) L49-52: You shouldn't number your listing if you never refer to it again. Furthermore, as stated above, this is a repetition of the part in the introduction on why landslide inventories are important and for what they are used.**

- We accept the suggestion of the referee.

- This paragraph has been removed from the new version of the manuscript.

**8) L95-111: I don't see how the meticulous listing of different methods on evaluating the completeness of landslide inventories is relevant for your work.**

- We believe that the list is preparatory to introduce our method that, in some way tackles the problem from another point of view. In this part of the manuscript, we list the existing methods proposed in the literature to evaluate the completeness of landslide inventories as a brief description of the state of the art.

- This part has been reduced and moved to the Introduction section as introductory information to support the justification and objectives of the work (lines 40-50 of the new version).

**9) L385: Here you generalize your findings to field surveys and even historical data as a whole, this does not follow, since you limited your investigation to field surveys "from roads".**

- We acknowledge that the sentence at L385 of the previous version of the manuscript can be understood as too much generic. Actually, what we wanted to explain here was the fact that among all the inventories tested in our study, just three showed a strong relation with the SA classes. So, in the new version of the manuscript we state that the proposed framework serves to test whether the information in existing inventories (especially those created by fieldwork) is affected by a scarce CoLM uniformity; and therefore, it enriches the portfolio of solutions to evaluate the quality of landslide inventories introducing the spatial component and local morphology in the analysis.

- We rewrote the Discussion and Conclusions section in order to better clarify this conclusion and avoid misleading generalizations.

**10) Bracketed citations should be at the end of a sentence not in the middle, if you need to refer to specific works within a sentence use in-text citation.**

- Right.

- The citations will be correctly positioned in the new version of the manuscript.

**Additional changes**

In addition to the changes related to the suggestions provided by the referees, we made some other relevant changes that, in our opinion, served to adjust in a better way the content of the manuscript. These changes are listed below:

- Title has been modified to "*Terrain visibility impact on the preparation of landslide inventories: a practical example in Darjeelig district (India)*"
- Former Figure 2 has been modified and moved. Now it is Figure 1.
- Former Figure 1 has been modified and moved. Now it is Figure 2
- Former Table 1 has been removed.
- Former Figure 3 has been modified. Now it incorporates also former Table 1.
- Former Table 2 has been modified. Now it is Table 1.
- Former Table 3 has been removed.
- Former Figure 4 has been modified.  Now it incorporates also former Table 3.
- Former Figure 5 has been modified.
- Former Figure 6 has been removed.
- Former Figure 9 has been modified. Now it is Figure 8.

---

## Author Response (AR2)

**Second Review Comments**
**nhess-2021-334**

We kindly appreciate the comments of the two referees and in the following lines we provide specific answers to their questions. In addition, where needed, we amended the manuscript in order to address the issues raised by the reviewers.

**Referee#1**

I believe that the manuscript can be published after minor revision. My specific comments are as follows:
1) Line 143 - 3.1 Data
- Please correct the section numbering. Section 3.1 was already used for Methods.
Thanks for pointing it out. We corrected the error.

2) Figure 3: Frequency area distribution curves (FAD curves) for Spot 2020 (green), GSI Historic (brown). Landslide distribution map for Spot 2020, GSI Historic, GSI Field and Gipuzkoa inventories.
- The Gipuzkoa inventory has been removed from Figure 3. It should also be removed from the figure caption.
Thanks for pointing it out. We corrected the error.

3) Figure 5:
- Please arrange the lithology types in the figure caption in the order in which they are given on the graph, starting with Brw.
Thanks for the suggestion. We corrected the order of the Land Use and Lithology types in the caption od Figure 5.

**Referee#2**
Stepping late into this review process, I think the submitted article is clearly within the scope of NHESS and a nice contribution regarding preparation and checking of landslide inventories. The paper has also clearly benefitted from previous revisions, and I think the article is not far from being acceptable for publication. While reading, I´ve made some minor observations that the authors should consider during final revision.

P2L63: CoLM: This is an important concept throughout the article and may be explained in more detail. Additionally, I am not sure if CoLM can be generally considered homogenous when using remotely sensed data because also here (e.g. for aerial photos) portions of the terrain can be covered by shadows or clouds and are hence not visible.
We thank the reviewer for this comment. We modified the manuscript in P2L60 as follows:
"*Also, the presence of shaded areas and clouds may hamper landslide recognition. However, for satellite or aerial images, the visibility of the territory is referred to the position of the sensor, and it can be assumed almost constant along the territory. Therefore, in this work we assume that inventories based in remotely sensed images were compiled in homogeneous working condition and then with uniform Capacity of Landslide Mapping (CoLM) over the studied area.*"

P3L87: "Sequences of different grades of rocks": What is meant here? Grades of metamorphism?
The reviewer is right. In this sentence "grades" refers to the grades of metamorphism. We modified the sentence in P3L85 as follows: "*The area is a part of active fold thrust belt of Darjeeling Himalayas where sedimentary rocks of Sub-Himalayas, low grade metamorphic sedimentary rocks of lesser Himalayas and high-grade metamorphic rocks of Higher Himalayas are present with or without the overburden cover of varied thickness. These sequences of different rocks are separated …*"
P4L105-108: The concept of EV should be illustrated with a suitable sketch to make it better understandable
Following the suggestion of the reviewer, we added an additional figure in the supplementary material. And we added the following text to the manuscript in P4L109 *"(Fig. A1 in the supplementary material illustrates the concept of EV and solid angle)"*
P6L145: GSI Field-inventory: What was mapped here? Generic landslides? Is there any typological information or are really just co-ordinates available?
There is no typological classification for this inventory. We only had the points positions.

P6L160: Isn´t it that the SPOT-inventory represents landslides originated in the same time span as those of the GSI Field-inventory? Maybe these inventories should be compared?

> Correct, indeed we initially checked the possible overlay between the SPOT and GSI-Field inventories, but it was not possible to recognize field data from the satellite images. As a consequence, they were used as two distinct inventories in the analysis.

P8L189: Landslide density in SA-classes is not shown in Fig. 5 but in Fig. 6?

> Correct, the sentence was misleading. We removed "see Fig. 5" because actually landslide densities are not shown in Fig. 5.

P11L236: Why was not the SPOT-inventory compared against the GSI-Field inventory in this manner? Both inventories aim surveying landslides of the same time period, so this might be of interest?

> Unfortunately, we don't have the information about the sizes of the landslides in GSI-Field, so the size distribution analysis can't be performed.

P12L250: What is the rationale here comparing a landslide inventory of a completely different area (with different characteristics in terrain visibility) with the Darjeeling inventories?

> We appreciate the comment of the reviewer and we added some lines in P12L256 in order to better explain the rationale. The added text is the following:
>
> "*We decided to run an extra experiment with this data set because we sought confirmation of the relevant role played by EV in influencing the spatial distribution of landslides in the inventories. In fact, since in Gipuzkoa information on the detailed road route followed by the surveyor was available, we expected the distribution of landslides to be even more influenced by visibility than in Darjeeling, where, due to the absence of specific information, the simulation of visibility was done using the entire road network. This additional inventory includes 542 shallow landslides and is referred to as* **Gipuzkoa** *inventory.*"

P13L287: But the roads in your study area are mostly located on ridges (as stated before) and hence might not be considered a causative factor for landslides since slope undercutting due to road construction might be excluded in your area?

> The reviewer is right, but in this part of the discussion we are addressing the general concepts about roads and landslide occurrence, referring to the literature. We are not referring to our concrete case in Darjeeling.

P14L324: The results obtained from the Spanish inventory cannot be verified by the reader since no EV class map or any other results from the visibility analyses are shown. I am not sure if introducing this inventory in your paper is helpful.

> Following the suggestion of the reviewer, we added an additional figure in the supplementary material. And we added the following text to the manuscript in P12L367 *"(Fig. A2 in the supplementary material shows the EV map obtained from the visibility analysis in Gipuzkoa)"*.